# Advantages of Ferroelectrics as a Component of Heterostructures for Electronic Purposes: A DFT Insight

**DOI:** 10.3390/ma16206672

**Published:** 2023-10-13

**Authors:** Irina Piyanzina, Alexander Evseev, Kirill Evseev, Rinat Mamin, Oleg Nedopekin, Dmitrii Tayurskii, Viktor Kabanov

**Affiliations:** 1Institute of Physics, Kazan Federal University, 420008 Kazan, Russia; i.piyanzina@gmail.com (I.P.); alexander-alexandrovich-evseev@mail.ru (A.E.); nedopekin@gmail.com (O.N.); dtayursk@gmail.com (D.T.); 2Zavoisky Physical-Technical Institute, FRC Kazan Scientific Center of RAS, 420029 Kazan, Russia; ekv97@mail.ru (K.E.); rf_mamin@yahoo.com (R.M.); 3Department for Complex Matter, Jozef Stefan Institute, 1000 Ljubljana, Slovenia

**Keywords:** ferroelectric, heterostructure, density functional theory, 2DEG, ME coupling

## Abstract

The main advantage of using ferroelectric materials as a component of complex heterostructures is the ability to tune various properties of the whole system by means of an external electric field. In particular, the electric field may change the polarization direction within the ferroelectric material and consequently affect the structural properties, which in turn affects the electronic and magnetic properties of the neighboring material. In addition, ferroelectrics allow the electrostriction phenomenon to proceed, which is promising and can be used to affect the magnetic states of the interface state in the heterostructure through a magnetic component. The interfacial phenomena are of great interest, as they provide extended functionality useful for next-generation electronic devices. Following the idea of utilizing ferroelectrics in heterostructural components in the present works, we consider 2DEG, the Rashba effect, the effect of magnetoelectric coupling, and magnetostriction in order to emphasize the advantages of such heterostructures as components of devices. For this purpose, model systems of LaMnO${}_{3}$/BaTiO${}_{3}$, La${}_{2}$CuO${}_{4}$/BaTiO${}_{3}$, Bi/BaTiO${}_{3}$, and Bi/PbTiO${}_{3}$, Fe/BaTiO${}_{3}$ heterostructures are investigated using density functional theory calculations.

## 1. Introduction

The presence of ferroelectric materials as component in heterostructures provides outstanding new functionality which can be used in electronic devices. It is well established that the appearance of a two-dimensional gas (2DEG) or liquid is possible due to the presence of internal electrical polarization. Such polarisation, for instance in LaAlO${}_{3}$/SrTiO${}_{3}$, arises due to the charge sequence in LaAlO${}_{3}$ atomic layers. However, 2DEG can occur even without charged atomic layers thanks to the presence of spontaneous polarisation in ferroelectric thin films [1,2,3,4,5,6]. This means that the electronic properties of the arising state can be tuned by an external field by changing the direction of the ferroelectric dipoles.

Another property which can be useful for electronic applications is magnetoelectric coupling. This property is associated with the possibility of controlling the ferromagnetic ordering at the interface due to interactions of spins through conduction electrons, leading to multiferroic properties on the part of the whole heterostructure. Multiferroic materials are compounds in which at least two order parameters coexist in the same phase. One very important and extremely rare group is that of ferroelectric ferromagnets, which have recently stimulated increasing research activity due to their scientific uniqueness and application in novel multifunctional devices. Magnetoelectric materials are mainly interesting due to the possibility of controlling the magnetic properties through an external electric field [7,8,9,10,11]. Due to the extraordinary challenges involved in creating multiferroic compounds, it is essential to create superlattice multicomponent materials, which consist of a magnetic insulator that supports spin-polarized 2DEG, and ferroelectric materials, which can aid in manipulating the magnetic state using an electric field, i.e., to realize a converse magnetoelectric (ME) effect [12].

Moreover, the presence of an electrostatic field in ferroelectrics due to spin–orbit (SO) coupling allows for control of SO splitting via ferroelectric polarization, which is a desirable property for spintronic applications. Natural materials demonstrating both the gigantic and ideal Rashba states are extremely rare [13,14]; computer simulations could help with this problem by allowing for the investigation of various combinations of materials.

A third promising effect for electronic devices purposes is magnetostriction in combination with ferroelectricity. In particular, within the superlattice approach it is possible to combine two materials with different features; by changing the linear sizes of the ferroelectric material by applying an external electric field the lattice parameters of a neighboring ferromagnet, the latter changes as well, leading to a change in the magnetic moments. This possibility promises significant advantages in the development of next-generation electronic devices. For instance, magnetic tunnel junctions (MTJs) are of great interest to both the experimental and theoretical communities due to their applications in magnetic random-access memory (MRAM) devices. Multiferroic materials are suitable for spin filter purposes as well [15,16]. Indeed, previous research has demonstrated that a heterostructure based on iron and a classical ferroelectric material (Fe/BaTiO${}_{3}$) can demonstrate ideal crystallinity and heteroepitaxial growth [17]. Even more significant is that changing the polarization direction is able to affect the magnetization inside a ferromagnetic film [8]. The Fe/BaTiO${}_{3}$ heterostructure has been widely investigated as a simple example of a model system of ferromagnetic/ferroelectric combination [5,8,17,18,19,20,21,22].

Within the last thirty years, a significant breakthrough in computational methods has been achieved thanks to the success of computer sciences. This has made possible various calculations of electronic and magnetic properties of sufficiently large and complex systems. In particular, the popular density functional theory has been implemented in a wide range of codes. Indeed, the possibilities of the approaches based on this method are very extensive.

Therefore, the present research is dedicated to an ab initio study heterostructures having a ferroelectric material as one of the components within the DFT approach. Our aim is to investigate the arising electronic and magnetic states and the possibilities for controlling the interfacial properties (2DEG, Rashba effect, ME coupling, magnetostriction) via ferroelectric polarization reversal. For this purpose, in this research we investigated LaMnO${}_{3}$/BaTiO${}_{3}$, La${}_{2}$CuO${}_{4}$/BaTiO${}_{3}$, Bi/BaTiO${}_{3}$, Bi/PbTiO${}_{3}$, and Fe/BaTiO${}_{3}$ heterostructures in order to demonstrate the effect of polarization switching onto the electronic and magnetic states, as well as the Rashba effect.

## 2. Materials and Methods

In the present research, calculations of structural, electronic, and magnetic properties were realized within the framework of density functional theory [23]. Exchange and correlation effects were accounted for using generalized gradient approximation (GGA-PBE) [24]. The Kohn–Sham equations [25] were solved using projectively extended wave potentials and wave functions [26]. All calculations were carried out using the VASP-6.3 (Vienna Ab initio Simulation Package) program [27] built into the MedeA computational software [28]. The cut-off of the plane wave was taken to be 400 eV, the convergence criterion for atomic relaxation was 0.02 eV/Å, and the convergence condition for self-consistent calculations was the invariance of the total energy of the system, with an accuracy of 10${}^{-5}$ eV. The Brillouin zones were sampled using Monkhorst–Pack grids [29,30,31], including 7 × 7 × 1 and 5 × 5 × 1 **k**-points, depending on the particular heterostructure under study. The Gaussian smearing was 0.05 eV. A set of calculations was carried out with a simplified *+U* correction applied [32], which was used to ensure a better description of the electronic properties of strongly correlated electrons; an additional *U* value was applied to electrons of the *d* and *f* orbitals, following [33]. In particular, we applied *U* = 4.4, 4, 4, 4.6 eV for 3*d* orbitals of Ti, Cu, Mn, and Fe, respectively, and 8 ev for 4*f* orbitals of La. The use of additional correction is essential to ensure a better description of the magnetic state and band gap. The necessity of adding the *U* parameters while treating the transition metals has been widely discussed in previous works, for example [33,34,35]. In addition, La 4*f* states are usually shifted up in the energy scale [35].

The model of the heterostructures was constructed in such a way that the BTOs served as an overlayer for LaMnO${}_{3}$/BaTiO${}_{3}$, La${}_{2}$CuO${}_{4}$/BaTiO${}_{3}$, and LaMnO${}_{4}$/BaTiO${}_{3}$; a vacuum region was added in order to imitate real heterostructures with both interface and surface regions. In the Bi/BaTiO${}_{3}$ and Bi/PbTiO${}_{3}$ heterostructures, the results are presented for structures with no vacuum region in order to provide a comparison with previous research [36]. Lastly, the model of Fe/BaTiO${}_{3}$ was the same as in [18], that is, a superlattice with no added vacuum, in order to focus on the interface and avoid surface impact. It should be noted that before investigating the interface, the energy of formation was checked in order to deal with the most stable among possible surface stakings. The all-slabs terminations at the interface were the most stable among those possible.

## 3. Results

In this section, the effects of the presence of ferroelectric material on the interfacial conducting state, the magnetic state, the size of Rashba-type splitting, and the reverse magnetostriction and magnetoelectric couplings are presented separately.

### 3.1. 2DEG

The area of perovskite-based heterostructures was investigated based on the appearance of a two-dimensional conducting state (2DEG) at the interface. The conductivity at the interface occurs due to either the polar nature of one of the components or to the presence of defects [37]. Later, was been shown that 2DEG can be created at the interface of non-polar oxides, one of which is ferroelectric [1]. The main advantage of using ferroelectrics is the possibility of switching the polarization on and off, allowing for control of the properties of the electron system. Moreover, ferroelectrics have a range of other outstanding properties which might expand the scope of applications in nanoscale electronic devices, including spontaneous polarization switching, high dielectric permeability, dielectric nonlinearity, piezo- and pyro- activity, and linear and quadratic electro-optical effects.

There are two systems studied in this frame within the present paper: the heterostructure of antiferromagnet/ferroelectric, i.e., LaMnO${}_{3}$/BaTiO${}_{3}$ (LMO/BTO), and ferroelectrics with high-temperature superconductors, such as as La${}_{2}$CuO${}_{4}$/BaTiO${}_{3}$ (LCO/BTO). Indeed, the creation of a 2DEG is possible when the electrostatic field along the slab is present in the system. This is possible due to the alternating charges in the atomic layers or thanks to the ferroelectric polarization being directed normally to the interface plane.

In both studied systems, the bulk components are insulators; LMO is A-type antiferromagnetic, LCO is a ferromagnetic insulator, and BTO is an insulator [38,39].

The unit cell for the LCO/BTO heterostructure is presented in Figure 1a, which consists of a middle slab of LCO as a substrate and BTO as an overlayer on both sides to make the cell symmetrical with respect to the central layer. It can be seen from the density of states (DOS) in Figure 1d that a conduction state arises, which is provided mostly by oxygen-polarized electrons. These conducting electrons are located mostly within the interfacial CuO layer (Figure 1b,c). In fact, the DOS at the Fermi level gradually increases from the surface of the ferroelectric towards the interfacial CuO layer and monotonously decreases towards the center of the LCO slab. The calculated number of the charge carrier dependence is rather discrete when carriers are present only within one atomic layer, indicating a 2D conducting character.

The other investigated heterostructure was LMO/BTO; it was constructed in the same way, with the antiferromagnet LMO surrounded by BTO overlayers on both sides, as depicted in Figure 2a. The optimization of the cell led to structural distortions associated with buckling within atomic layers. The most pronounced displacements were found close to the interface. These out-of-plane shifts contributed to potential build-up along the BTO overlayer, giving rise to the internal field; however, the resulting electrostatic field was not sufficiently large to promote significant electronic reconstruction and conductivity in the system, as can be seen in Figure 2d, where the DOS at the Fermi-level is zero.

In the same way, increased polarization due to the artificial displacement of positive ions with respect to the negative oxygen ions immediately results in increased charge carriers at the surface (holes) and interface (electrons), as depicted in Figure 2b. In terms of energy state, the increased polarization shifts the Fermi level upwards (Figure 2d,e), meaning that the Ti and Mn 3*d* states become closer to the Fermi level. Contrarily, polarization towards the interface leads to an opposite situation, with holes located near the interface and electrons located near the both surfaces, as depicted in Figure 2c. The presence or absence of conducting state is mentioned in Table 1.

To sum up, the presence of the electrostatic field of the ferroelectric material is favorable in the systems where 2DEG is a desirable property. There are at least two advantages of such components: first, the arising conducting phase might be switched by external field stimulus, and second, the field within the ferroelectric material is an intrinsic feature, meaning that the requirement of an ideal interface is not indispensable here, which makes growth easier. Consequently, both features are preferred for electronic purposes.

### 3.2. Reverse Magnetoelectric (ME) Coupling

As mentioned in the introduction, multiferroics are of a great scientific interest due to the wide range of their physical properties. Furthermore, this class of materials has a great potential for applications in switches, magnetic field sensors, and memory devices [40]. However, pure multiferroics are rare, necessitating searching among those multicomponent superlattices mainly constructed from ferroelectrics and magnets.

In the previous section, it was clearly demonstrated that the change in ferroelectric polarization direction switches the conductivity on and off. To test the possibilities, the distribution of magnetic moments within the antiferromagnetic slab of LMO was checked. It has to be noted that while LMO is a pure antiferromagnet in the bulk, due to the odd number of five MnO layers in the slab in the slab geometry used in the present work, the order is that of an unsaturated antiferromagnetic in total. This is different from the situation described in [4], for instance, where an infinite cell without a vacuum region was used for investigation. In the model constructed within the present research, the effect of changed polarization direction was observed. As listed in Table 1, the initial optimized LMO/BTO heterostructure possesses alternating magnetic moments directed along and opposite to the *z*-axis. This order is preserved for other considered cases as well, however, the magnitude changes. While these changes are insignificant, the situation might change in the superlattice geometry, which needs to be checked in further investigations.

To conclude, the ab initio observation of magnetic moments switching through the reverse magnetoelectric coupling requires an adjustment to the geometry of the heterostructure. In particular, in the LMO/BTO structure considered here, the change in polarization direction does not change the direction of the magnetic moments of Mn ions, though it does change the amplitude.

### 3.3. Rashba Effect

The Rashba effect constitutes the splitting of the electron conduction band along the spin due to the spin–orbit interaction. The effect is observed in structures where an effective electric field is present. Due to the presence of spin-orbit interaction, these internal electric fields lead to a splitting of the electronic states along the axis of the wave vectors. As a result, two dispersion surfaces are formed, which are connected at one Dirac point. A large and ideal Rashba-type splitting is desirable for applications in spintronic devices. There are a number of approaches that exist for enhancing this splitting. In particular, introduction of heavy elements as components of heterostructure may lead to an increase in the strength of the spin–orbit (SO) coupling [41]. Another approach is to use polar semiconductors as a substrate for the heterostructure [42,43]. This avoids mixing of the Rashba states and the spin-degenerate substrate states, allowing for the creation of so-called ideal Rashba states. The last and most promising approach is to use ferroelectric materials to enhance the electric field across the heterostructure [44]. Furthermore, the use of ferroelectrics allows for tuning the polarization, which in turn may lead to changes in the strength of SO coupling.

In the present research, two heterostructures were considered, BaTiO${}_{3}$/Bi and PbTiO${}_{3}$/Bi, both consisting of a ferroelectric substrate and heavy metal monolayer. The BaTiO${}_{3}$/Bi heterostructure was investigated previously in [36,45]. Here, we followed the same heterostructure model except with the addition of a vacuum region. The structures of modeled BTO/Bi and PTO/Bi cells, along with corresponding band structures, are presented in Figure 3. One band with splitting is presented, which is the same for the S${}_{x}$ and S${}_{y}$ components; as the out of plane component S${}_{z}$ is negligibly small, the spins are located and split within the interfacial plane. Two path directions in the Brillouin Zone were found, namely, X–G and G–M, where G is a gamma point.

Dispersions of surface states with characteristic features of Rashba-type splitting (shifting of the energy E${}_{R}$ and Rashba wave vector k${}_{0}$) were found for both the BaTiO${}_{3}$/Bi and PbTiO${}_{3}$/Bi heterointerfaces. All data are collected in Table 2 along with effective masses and ab initio data from [45].

All obtained results for Bi/BTO are in qualitative agreement with previously published data; the difference might be due to differences in the computational parameters. The replacement of BTO by PTO increases the Rashba parameter by a factor of ≈1.7, which is a significant increase. Indeed, in PTO the ferroelectric polarization associated with displacements of positively charged Ti ions out of negatively charged oxygen planes is significantly higher, as can be seen in Figure 3. Thus, the electrostatic field along the cell and perpendicular to the interface plane is higher, leading to more splitting.

Further, the Bi monolayer was checked separately in order to ensure that the splitting occurs only when both spin–orbit coupling and a field perpendicular to the interface plane coexist in the heterostructure. The Bi monolayer with corresponding band structure is presented in Figure 4, where the splitting is present but is not of Rashba-type.

Finally, we checked the influence of applying a reverse field by shifting in the opposite direction with respect to the oxygen planes and freezing the atoms in the ferroelectric slab. This resulted in similar values to those listed in Table 2. However, the increase in ferroelectric polarization might increase the splitting. This could be a subject of further investigations, along with other combinations of heavy elements and different ferroelectrics.

### 3.4. Reverse Magnetostriction Effect

We chose the Fe/BaTiO${}_{3}$ model system was to study the effect of linear compression sizes of the ferroelectric material on the magnetic properties of the ferromagnet. The unit cell used in the calculations is depicted in Figure 5a, where BTO acts as a ferroelectric and iron acts as a ferromagnetic overlayer. In this case, the unit cell was constructed without a vacuum region in order to exclude the impact of the surface and concentrate on the ferroelectric slab properties. The system was constructed to include a total of eighteen atomic layers in the periodic structure, seven of which were layers of bcc iron.

The initial value of the lattice parameter was equal to 3.9 Å, corresponding to the BTO substrate (the lattice mismatch was 1.36%). We calculated the values of the magnetic moments of iron atoms in the BTO/Fe heterostructure without striction, taking into account spin–orbit interactions; the data are shown in Figure 5b. After that, the entire heterostructure was simultaneously compressed along the *x*-axis by 0.2 Å and expanded along the *y*-axis by the same value. After an optimization process taking into account spin–orbit interactions, the magnetic moments of the iron atoms in each layer of the ferromagnet were calculated; these data are presented in Figure 5c. Taking spin–orbit interactions into account during the calculation makes it possible to obtain magnetic moment values that have different directions, and not just the total magnetization value of the Fe layers.

The Figure 5b shows that all curves are symmetrical with respect to the middle of the layer and have the same character of a slight increase near the interfaces. The maximum value of the total magnetic moment was found to be ≈3.22 $\mu_{B}$, which is 0.23 $\mu_{B}$, higher than the initial magnetic moment of iron (2.99 $\mu_{B}$) calculated per Fe ion. The magnetic moments are predominantly directed along the *x*- and *z*-axes; the magnitude of magnetization directed along the *z*-axis is located in the region from ≈1.94 $\mu_{B}$ to 2.04 $\mu_{B}$, along *x*-axis from ≈1.74 $\mu_{B}$ to 1.83 $\mu_{B}$, and along the *y*-axis from ≈1.61 $\mu_{B}$ to 1.69 $\mu_{B}$.

In the case of a heterostructure with compression of Fe/BTO, the magnetic moment curves have the same character. The maximum value of the total magnetic moment does not change, though the minimum value does; it is observed in layer 2 and is equal to ≈2.96 $\mu_{B}$. The locations of the magnetization distribution curves along the axes have changed; in the case of the applied striction, the *y*-axis becomes predominant, the values of which range from ≈1.83 $\mu_{B}$ to 1.99 $\mu_{B}$, while along the *z*-axis the values range from ≈1.72 $\mu_{B}$ to 1.89 $\mu_{B}$ and along the *x* axis—from ≈1.56 $\mu_{B}$ to 1.70 $\mu_{B}$.

In order to check whether the change in magnetization is due to the shape change of BTO and corresponding influence or just from the change of the lattice parameters of Fe, we carried out a similar spin–orbit calculation only for pure bulk iron. The structure of bulk Fe is shown in Figure 6a, and was constructed in such a way that its lattice parameters (*x* and *y*) are equal to the corresponding lattice parameters of BTO, mimicking the substrate conditions. We used seven atomic layers of bcc iron, as in the heterostructure depicted in Figure 5. Figure 6b shows the results of the magnetization distribution over the atomic layers for that cell. We found all the distributions in Figure 6b to be constant, with the *x* and *y* components equal in terms of magnitude while the *z*-component is slightly higher (less than 0.005 $\mu_{B}$). The total magnetization is same as in the heterostructure (Figure 5). Then, the cell was compressed along the *x*-axis and extended along *y*-axis, as in the previous case. Similar to the heterostructure case, the magnetization vector turned towards the extended *y* direction, as depicted in Figure 6c. Consequently, the *z*-component now coincides with the *x* component and the *y* component is 0.01 $\mu_{B}$ higher.

Overall, in the case of the heterostructure the effect is more pronounced, the magnitudes of all components are similar, and the total magnetization is roughly the same; however, there are a few significant differences. First, the components are more distant from each other, meaning that it may be possible to distinguish them easier. In the heterostructure, the difference between the *x* and *y* components is ≈0.1 $\mu_{B}$, while the *z* component is higher by ≈0.2 $\mu_{B}$. The change in magnetic moments along the *y* direction is ≈0.3 $\mu_{B}$ due to extending the lattice by 0.2 Å. Moreover, this extension and compression lead to increments of difference in the *x* and *y* components from 0.1 $\mu_{B}$ to 0.4 $\mu_{B}$, whereas in the bulk the difference is only 0.01 $\mu_{B}$. Thus, our results show that compression along one axis and expansion along the other can affect the direction of magnetization such that the magnetic moment turns predominantly along the axis with expanded length. In the heterostructure geometry, this effect is significant and could find interesting practical applications.

## 4. Conclusions

In this paper, we have demonstrated that the incorporation of ferroelectrics as a component of the heterostructure may enhance the desirable properties of the heterostructure.

In the case of the creation of a interfacial conducting state, the presence of a material with intrinsic ferroelectric polarization is favorable due to the opportunity to toggle the conductivity by means of an external field. Using the example of two heterostructures, LaMnO${}_{3}$/BaTiO${}_{3}$ and La${}_{2}$CuO${}_{4}$/BaTiO${}_{3}$, we have seen that even without charged layers, as in LaAlO${}_{3}$/SrTiO${}_{3}$, the 2DEG might arise within the interfacial layers. Furthermore, another advantage concerns the growth conditions of the ferroelectric overlayer. Indeed, the alternation of charges in LaAlO${}_{3}$/SrTiO${}_{3}$-type heterostructures and the pure interface without defects are substantial conditions for a 2DEG, as the field arises in the ferroelectric material anyway.

The second very important feature desirable for electronic applications, especially for memory devices, is reverse magnetoelectric coupling. Systems with the ability to change the magnetic ordering by means of electric stimulus are of particular interest due to higher energy storage density. In the present paper, we observed the effect of the ferroelectric polarisation on the magnitude of magnetic moments in the LaMnO${}_{3}$/BaTiO${}_{3}$.

The presence of ferroelectric material as a component of a heterostructure along with a heavy metal such as Bi may enhance splitting thanks to ferroelectric polarization. In addition, the external electric field may increase the internal electrostatic field and slightly increase the splitting. The change in the ferroelectric polarization direction does not significantly change the Rashba parameter.

Lastly, the impact of anisotropic striction of ferroelectric BaTiO${}_{3}$ on the magnetic moments of ferromagnetic bcc iron was investigated. Striction in one direction (*x*-axis) and corresponding extension in the other direction (*y*-axis) resulted in magnetization redistribution; in particular, the *y* component of the magnetization vector becomes predominant, whereas in the initial heterostructure the *z* component has the highest magnitude. Furthermore, comparison to bulk Fe with same lattice parameters suggests the conclusion that BTO has additional impact on the distribution of the magnetic moments. More precisely, the magnetization vectors in the heterostructure components are more pronounced, i.e., there is a ≈0.2 $\mu_{B}$ difference between each component without applied striction; moreover, the applied anisotropic striction makes the difference between the *x* and *y* components equal to 0.3 $\mu_{B}$, which is significant and might find interesting applications.

Overall, the use of ferroelectrics opens up new possibilities when combining materials with different properties within one device, and can make possible to tune certain properties of the interfacial state, namely, conductivity, Rashba splitting, and magnetization. This can provide an opportunity to tune the magnetic moments via the magnetoelectric coupling and striction. The present research provides an overview of areas where ferroelectrics might be useful and should be considered as possible components in a multilayer structure. In other words, it provides examples of phenomena which arise in heterostructures with ferroelectrics and which might be promising avenues for further investigation. Lastly, the research presents a general methodology used for treating heterostructures via ab initio calculations.

## Figures and Tables

**Figure 1 materials-16-06672-f001:**
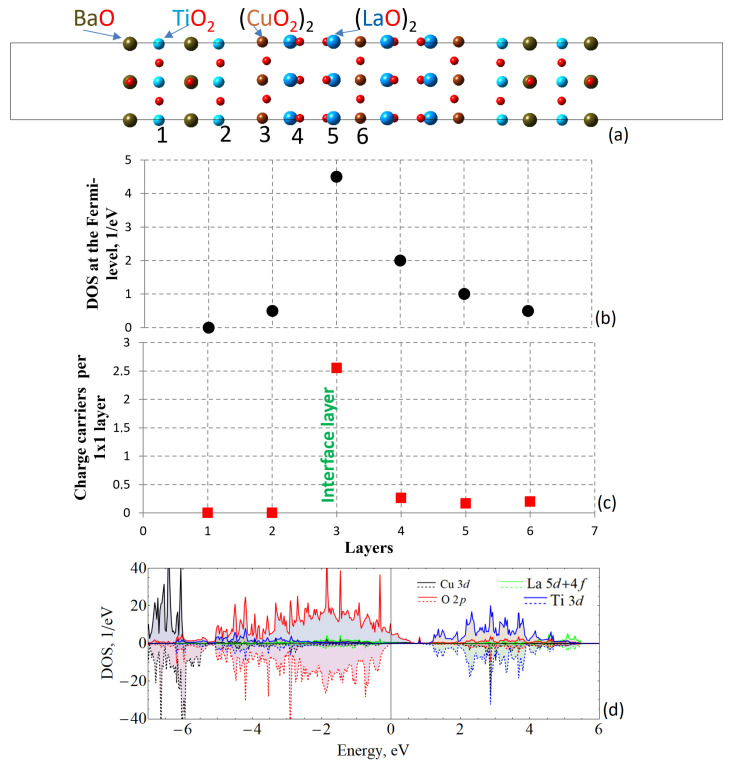
(**a**) The unit cell of the La${}_{2}$CuO${}_{4}$/BaTiO${}_{3}$ (LCO/BTO) heterostructure; (**b**) density of states (DOS) per atomic layers as denoted in (**a**); (**c**) charge carriers per 1 × 1 layer; (**d**) atom-resolved DOS, where the spin components are presented at the upper and bottom parts of the graph. The red squares and black dots correspond to charge carriers and DOS at the Fermi-level respectively.

**Figure 2 materials-16-06672-f002:**
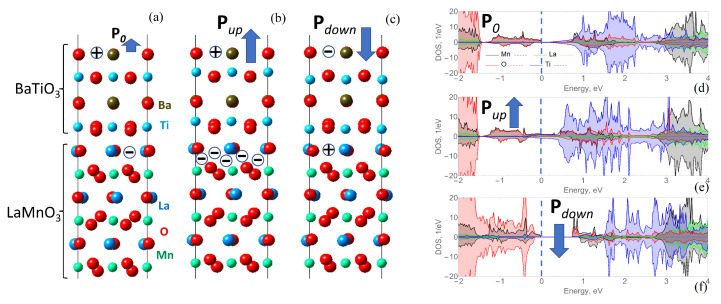
Half unit cells of the LaMnO${}_{3}$/BaTiO${}_{3}$ (LMO/BTO) heterostructure (**a**) fully optimized and with imposed polarization directed (**b**) towards the surface and (**c**) towards the interface (**d**–**f**) with corresponding atom-resolved density of states (DOS), where the spin components are presented at the upper and bottom parts of the graph. **P**${}_{0}$ denotes the initial polarization, **P**${}_{up}$ is the polarization directed towards the surface, **P**${}_{down}$ is the polarization directed towards the surface the interface, and the plus and minus symbols respectively correspond to the positive and negative charge distributions.

**Figure 3 materials-16-06672-f003:**
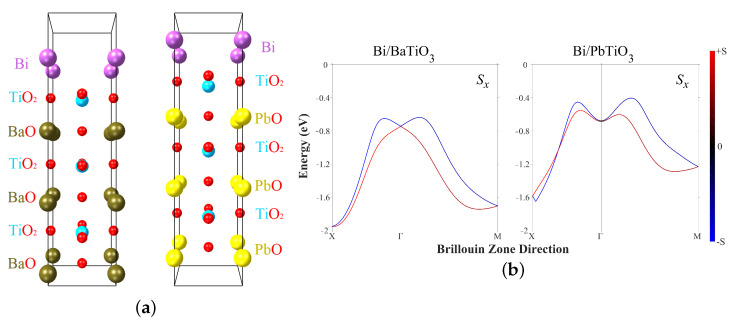
(**a**) The unit cells and (**b**) the corresponding band structures of the investigated BTO/Bi and PTO/Bi heterointerfaces. Only the S${}_{x}$ component of the split band is presented here, as the S${}_{y}$ component is the same and the S${}_{z}$ component is zero. The red and blue colors correspond to the spin-up and spin-down components.

**Figure 4 materials-16-06672-f004:**
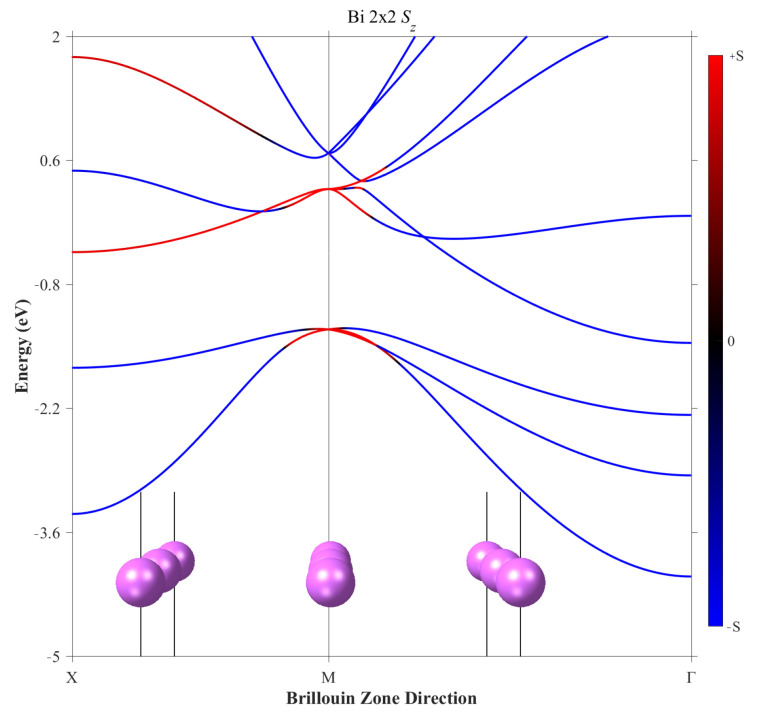
The unit cells and corresponding band structure of a 2 × 2 Bi mono-layer with no Rashba-type splitting. The red and blue colors correspond to the spin-up and spin-down components.

**Figure 5 materials-16-06672-f005:**
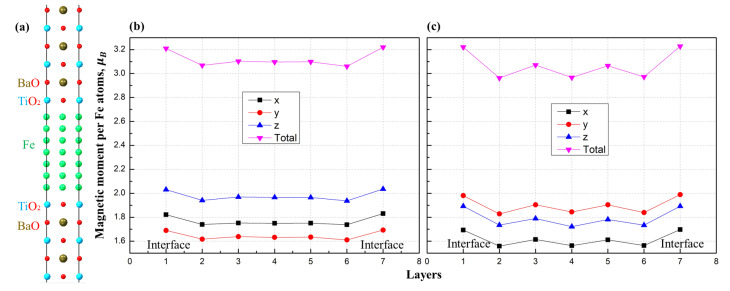
(**a**) The unit cell of Fe/BaTiO${}_{3}$ heterostucture used in calculations: (**b**) magnetic moments of Fe atoms in the heterostructure without applied striction and (**c**) corresponding distribution of the magnetic moments of Fe atoms calculated for each layer of Fe/BTO heterostructure within the applied in-plane striction along the *x*-axis and extension along the *y*-axis. Each curve corresponds to the resulted magnetisation magnitude for the *x*, *y*, and *z* components of the magnetization vector and total magnetic moment in the scalar norm.

**Figure 6 materials-16-06672-f006:**
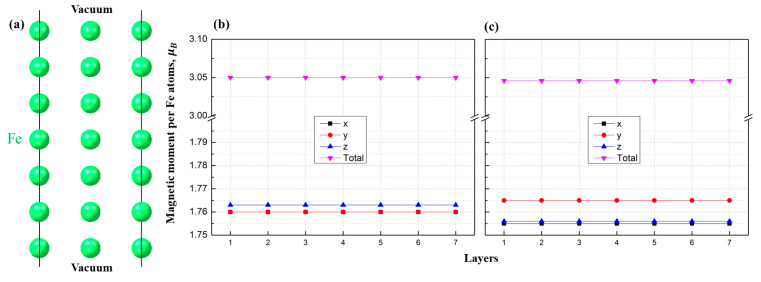
(**a**) The unit cell of the Fe structure used in the calculation, (**b**) the corresponding distribution of the magnetic moments of Fe atoms calculated for each layer of the structure with in-plane striction, as was in the case for the BTO substrate (1.36% mismatch), and (**c**) the corresponding distribution within the applied in-plane striction along the *x*-axis and extension along the *y*-axis in accordance with Figure 5. Each curve corresponds to the resulting magnetisation magnitude for the *x*, *y*, and *z* components of the magnetization vector and total magnetic moment in the scalar norm.

**Table 1 materials-16-06672-t001:** Conductivity and distribution of magnetic moments per Mn ions over atomic layers within the LaMnO${}_{3}$ slab of the LaMnO${}_{3}$/BaTiO${}_{3}$ heterostructure; the + and − symbols in the conductivity row respectively denote the presence and absence of the conducting state in the investigated systems.

	P${}_{0}$	P${}_{\mathrm{down}}$	P${}_{\mathrm{up}}$
Conductivity	−	−	+
Interfacial layer	−3.725	−4.165	−3.710
Second layer	3.670	3.635	3.705
Middle layer	−4.170	−3.91	−3.710
Second layer	3.675	3.635	3.710
Interface layer	−3.725	−4.165	−3.630

**Table 2 materials-16-06672-t002:** Calculated Rashba splitting parameters, where E${}_{R}$ is the Rashba energy, k${}_{0}$ is the momentum offset, α${}_{R}$ is the Rashba parameter, m* is the effective mass, and X–G and G–M denote the path in the Brillouin Zone. The results are shown for both fully optimized heterostructures of BaTiO${}_{3}$/Bi and PbTiO${}_{3}$/Bi with optimized polarization directed towards the interface.

Path	X–G				G–M			
	**E${}_{R}$, eV**	**k${}_{0}$, Å${}^{-1}$**	**α${}_{R}$, eVÅ**	**m*, m${}_{e}$**	**E${}_{R}$, eV**	**k${}_{0}$, Å${}^{-1}$**	**α${}_{R}$, eVÅ**	**m*, m${}_{e}$**
Bi/BTO	0.1	0.19	1.05	1.35	0.11	0.22	1.01	1.65
Bi/PTO	0.23	0.27	1.72	1.2	0.28	0.35	1.63	1.61
Bi/BTO [45]	0.16	0.22	1.45	1.14	0.18	0.25	1.42	1.36

## Data Availability

The data presented in this study are available on request from the corresponding author.

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
