# Peer review of "Advantages of Ferroelectrics as a Component of Heterostructures for Electronic Purposes: A DFT Insight"

_materials, 2023, doi:10.3390/ma16206672_

Round 1
Reviewer 1 Report
The manuscript “materials-2648377” titled as “Advantages of ferroelectrics as a component of heterostructures for electronic purposes: a DFT insight” deals with the ab initio method to investigate the LaMnO3/BaTiO3, La2CuO4/BaTiO3, Bi/BaTiO3 and Bi/PbTiO3, Fe/BaTiO3 heterostructures. I have a few minor queries before accepting the manuscript.
1. The authors have used U = 4.4, 4, 4 eV for 3d orbitals of Ti, Cu and Mn and 8 ev for 4f orbitals of La, respectively. For Cu, the electronic configuration is 3d¹⁰ 4s¹. Therefore, the 3d orbitals are completely full. So, was it necessary so high values of U for Cu. It is very important to understand how they adapted these particular values of U for all the elements. How did the author choose these values? Similarly, why the U value for Mn is smaller than Ti? Provide insight. For Fe what is the value of U?
2. Did the authors consider the spin-orbit coupling? If not, then authors can check how the results are different when one considers spin-orbit coupling and mentioned in the manuscript.
3. Provide the spin configuration for different heterostructures?
4. What is the novelty of this work please mention it properly, since there are multiple heterostructures, the main research achievement is not very clear.
Author Response
Dear Referee, many thanks for your comments and suggestions! Please, find below our answers.
The manuscript “materials-2648377” titled as “Advantages of ferroelectrics as a component of heterostructures for electronic purposes: a DFT insight” deals with the ab initio method to investigate the LaMnO3/BaTiO3, La2CuO4/BaTiO3, Bi/BaTiO3 and Bi/PbTiO3, Fe/BaTiO3 heterostructures. I have a few minor queries before accepting the manuscript.
- The authors have used U = 4.4, 4, 4 eV for 3d orbitals of Ti, Cu and Mn and 8 ev for 4f orbitals of La, respectively. For Cu, the electronic configuration is 3d¹⁰ 4s¹. Therefore, the 3d orbitals are completely full. So, was it necessary so high values of U for Cu. It is very important to understand how they adapted these particular values of U for all the elements. How did the author choose these values? Similarly, why the U value for Mn is smaller than Ti? Provide insight. For Fe what is the value of U?
Indeed, the Cu orbitals are full. Research by Wang et all (Phys. Rev. B 73, 195107 https://doi.org/10.1103/PhysRevB.73.195107) has demonstrated that the use of additional correction is essential for a better description of the magnetic state and band gap. In particular, just GGA gives 0 for both magnetization and band gap according to that research. The other parameters were also taken from similar conclusions summarized in Ref. (http://dx.doi.org/10.1016/j.commatsci.2015.07.019).
We have forgotten to add U(Fe) = 4.6eV, which was also taken from the same research.
You are right, the choice of the U parameter is tricky and requires additional pre investigation. However, that is difficult and no one usually does that. In one of our early research, we already made an attempt to investigate the impact of U in the LAO/STO heterostructure (http://dx.doi.org/10.1088/1361-648X/aa57ac), where various U were applied for Ti and La states. In that study, we performed a systematic variation of the U − J values for the aforementioned two orbitals and their effect on the atomic and electronic structure, which led us to conclude that the above values represent good choices and thereby allow for an accurate and at the same time efficient assessment of the systems under study. This conclusion was also supported by one of the previous works using the hybrid functional approach [https://doi.org/10.1103/PhysRevB.88.045119]. We have also checked that for Ti there is no big difference in the band gap for U=2 and 4 eV.
We have added some explanation to the Method section.
- Did the authors consider the spin-orbit coupling? If not, then authors can check how the results are different when one considers spin-orbit coupling and mentioned in the manuscript.
Yes, we took spin-orbit coupling into account in the Fe/BTO calculations to investigate the change in spatial distribution during applied striction and for the heterostructure without applied striction, the results are shown in section 3.4. Indeed, the result has changed, since the spin-polarized calculations that we had in previous works show only the total magnetic moment of iron atoms, whereas now we have obtained the distribution of magnetic moments in the iron layers in the BTO/Fe heterostructure along all axes.
- Provide the spin configuration for different heterostructures?
For the heterostructures with Rashba splitting we plotted the heatmaps on the right sides of each Band structure plot, namely, Figure 3b, 4b with red and blue colors corresponding to spin configurations. We have added that to the caption. The other heterostructures as LMO/BTO and LCO/BTO were analyzed by densities of states. The upper and lower parts correspond to different spin configurations respectively. We added that in the caption as well. The last Fe/BTO was calculated within the spin-orbit approach, so spins were considered degenerated, only spatial distribution was investigated here.
- What is the novelty of this work please mention it properly, since there are multiple heterostructures, the main research achievement is not very clear.
Well, we have collected our recent calculations in one summary in order to emphasize the idea of using the ferroelectrics within the heterostructure, to show the whole methodology mostly used while dealing with heterostructures.
Most results were updated with higher accuracy or new approaches. In particular, for LMO/BTO case an impact of ferroelectric polarization onto the magnetic moments was checked, for Bi/BaTiO3 and Bi/PbTiO3 we used other geometry which resulted in higher Rashba-splitting, and the comparison with Bi monolayer itself was performed in order to proof the source of splitting. Then, for Fe/BaTiO3 the spin-orbit coupling was taken into account and the compression along with one axis was checked, a comparison with other geometries of pure Fe under the compression was performed, namely in the bulk forms. We gave some generalized conclusions about the advantages of using ferroelectric as a component of heterostructure. We tried to add a more precise conclusion at the end.
Reviewer 2 Report
Gumarova et. al. presented a DFT study of several heterostructures involving ferroelectric materials in this paper. They discussed the effect of the ferroelectric polarization of BTO on the 2DEG on the interface between LMO/BTO and LCO/BTO, on the magnetic moments in Mn ions in LMO/BTO heterostructure, and on the Rashba effect in Bi/BTO and Bi/PTO, as well as the effect of strain on the magnetic properties of Fe in Fe/BTO.
The scientific methodology and results appear generally solid and comprehensive. Nevertheless, most of the results add little to the previous literature and the only new things, in my opinion, are the calculations of LCO/BTO and Fe/BTO, which consist only a small portion of the paper. In addition, many presentations and figures are not clear enough, and many calculations lack proper analysis on top of direct numerical results. Consequently, with the reasons detailed below, I do not recommend publishing this paper, unless the authors address properly my concerns and make major revisions accordingly.
Major questions and concerns:
1. novelty: for 2DEG discussion, most findings were reported in J. Mater. Sci. (2022) 57:21620–21629 from some of the same authors. For Rashba effect, the difference with Ferroelectrics 605.1 (2023): 27-35 is simply the choice of vacuum layers (see Lines 191-193, 203-204 by the authors themselves). Most of the paper simply confirmed previous research with minor differences.
2. About the general method employed in the magnetostriction of Fe/BTO (Figure 5): the authors achieved magnetostriction by "the applied in-plane striction along x-axis and extension along y-axis." Nevertheless, this will also change the lattice constant of Fe, whose magnetic properties, even without BTO, may change. I could not see how the authors are sure that the change in magnetization is from the shape change of BTO. In addition, the word "magnetostriction" means "change of shape or dimensions during the process of magnetization", which is different than what the authors described.
Some minor questions:
1. The LCO/BTO heterostructure presented in Figure 1 (a) shows interface between TiO plane and CuO plane. Is this the only possibility, or can the interface be between, for example, CuO plane and BaO plane? Would the result change?
2. Line 143: "located near the surface", which surface, top or bottom?
3. In Figure 2 (a-c), the atoms are not labeled
4. Line 161: odd layers of AFM material are not "ferromagnetic", because the coupling between adjacent layers is still anti-ferromagnetic. Instead, I would suggest using the expression "unsaturated AFM".
5. Table 1: what does - and + mean for conductivity? Why are there two interface layers and two second layer? There is no informative caption nor discussion in the main text, which is confusing.
6. Line 213: "splitting is present but not a Rashba-type." What splitting? If not Rashba, what is the origin?
7. Line 216: "That resulted in similar values as listed in Table 2." Does this mean that reverse field changes nothing?
8. Line 239: "The magnetic moments are predominantly directed along the y and y axes". This is contradictory to Figure 5(b).
Some grammatical details:
1. Line 149: "privilege" is not the right word
2. Line 177: replace "presents" by "is present"
Author Response
Dear Referee, many thanks for your comments and suggestions! Please, find below our answers.
Gumarova et. al. presented a DFT study of several heterostructures involving ferroelectric materials in this paper. They discussed the effect of the ferroelectric polarization of BTO on the 2DEG on the interface between LMO/BTO and LCO/BTO, on the magnetic moments in Mn ions in LMO/BTO heterostructure, and on the Rashba effect in Bi/BTO and Bi/PTO, as well as the effect of strain on the magnetic properties of Fe in Fe/BTO.
The scientific methodology and results appear generally solid and comprehensive. Nevertheless, most of the results add little to the previous literature and the only new things, in my opinion, are the calculations of LCO/BTO and Fe/BTO, which consist only a small portion of the paper. In addition, many presentations and figures are not clear enough, and many calculations lack proper analysis on top of direct numerical results. Consequently, with the reasons detailed below, I do not recommend publishing this paper, unless the authors address properly my concerns and make major revisions accordingly.
- novelty: for 2DEG discussion, most findings were reported in J. Mater. Sci. (2022) 57:21620–21629 from some of the same authors. For Rashba effect, the difference with Ferroelectrics 605.1 (2023): 27-35 is simply the choice of vacuum layers (see Lines 191-193, 203-204 by the authors themselves). Most of the paper simply confirmed previous research with minor differences.
Well, we have collected our recent calculations in one summary in order to emphasize the idea of using the ferroelectrics within the heterostructure, to show the whole methodology mostly used while dealing with heterostructures.
Most results were updated with higher accuracy or new approaches. In particular, for LMO/BTO case an impact of ferroelectric polarization onto the magnetic moments was checked, for Bi/BaTiO3 and Bi/PbTiO3, we used other geometry which resulted in higher Rashba-splitting, and the comparison with Bi monolayer itself was performed in order to proof the source of splitting. Then, for Fe/BaTiO3 the spin-orbit coupling was taken into account and the compression along with one axis was checked, a comparison with other geometries of pure Fe under the compression was performed, namely in the bulk forms. We gave some generalized conclusions about the advantages of using ferroelectric as a component of heterostructure. We tried to add a more precise conclusion at the end.
2.About the general method employed in the magnetostriction of Fe/BTO (Figure 5): the authors achieved magnetostriction by "the applied in-plane striction along x-axis and extension along y-axis." Nevertheless, this will also change the lattice constant of Fe, whose magnetic properties, even without BTO, may change. I could not see how the authors are sure that the change in magnetization is from the shape change of BTO. In addition, the word "magnetostriction" means "change of shape or dimensions during the process of magnetization", which is different than what the authors described.
Thank you for bringing this to our attention. We have included in the revised version of the article the properties of the pure iron slab without BTO under the same striction condition as for the studied BTO/Fe heterostructure. We considered the same number of layers and the same lattice parameters. The magnetic properties of such compressed iron differ from our results in the case of the BTO/Fe heterostructure. That means that there is an interaction leading to different properties and features. We have added an additional paragraph to the 3.4 Section.
In particular:
indeed, most features come due to compression. That was expected since we considered a real situation when the compression of BTO due to the electrostriction forces the neighboring material, here Fe, to compress as well.
However, the presence of BTO changes the magnetization not only through the change of lattice, but through other mechanisms, possibly magnetoelectric coupling.
You are right about our statements regarding the effect of striction on the magnetic properties of the BTO/Fe heterostructure, since we did not give an example of a calculation of a heterostructure without striction, thank you for pointing this out. We corrected this; a graph of magnetization of the structure BTO/Fe without striction was added to the text and it is Figure 5b.
You are also right about the proper use of the word “magnetostriction”. Because the object of our study is reverse magnetostriction. By applying compression we study the change of magnetization in the ferromagnetic slab. We changed this for the final version of paper
Some minor questions:
- The LCO/BTO heterostructure presented in Figure 1 (a) shows interface between TiO plane and CuO plane. Is this the only possibility, or can the interface be between, for example, CuO plane and BaO plane? Would the result change?
Indeed, 4 possible combinations might exist. Every time, before constructing the interface, the energy of formation is checked in order to deal with the most stable one. Besides, most of the time only one (or two sometimes) heterointerface is possible to converge within the optimization procedure. We have added a sentence in the Method section concerning that point.
- Line 143: "located near the surface", which surface, top or bottom?
Within our research for LMO/BTO a supercell with a central slab of LMO and two BTO overlayers on both sides of the LMO was used. The cell has a mirror symmetry with respect to the central layer of LMO. In such geometry both surfaces (and interfaces) are identical and the conclusions made here are true for both surfaces and interfaces.
- In Figure 2 (a-c), the atoms are not labeled
We added the labling.
- Line 161: odd layers of AFM material are not "ferromagnetic", because the coupling between adjacent layers is still anti-ferromagnetic. Instead, I would suggest using the expression "unsaturated AFM".
Many thanks, we have changed to that expression.
- Table 1: what does - and + mean for conductivity? Why are there two interface layers and two second layer? There is no informative caption nor discussion in the main text, which is confusing.
Indeed, we have added the caption and description in the text.
- Line 213: "splitting is present but not a Rashba-type." What splitting? If not Rashba, what is the origin?
Thanks for your question. At the moment, it has only been determined that the bismuth monolayer does not have Rashba-type splitting. But the nature of the origin of dispersion lines similar to splitting in this band structure has not yet been determined.
- Line 216: "That resulted in similar values as listed in Table 2." Does this mean that reverse field changes nothing?
Yes, indeed, the application of a reverse field in a given heterostructure does not significantly change the value of the Rashba parameter.
- Line 239: "The magnetic moments are predominantly directed along the y and y axes". This is contradictory to Figure 5(b).
Thank you for your comment. We made a mistake, we should have written "The magnetic moments are predominantly directed along the y and z axes". In the revised version of the text, we have corrected this.
Comments on the Quality of English Language
Some grammatical details:
- Line 149: "privilege" is not the right word
It has been changed to “preferred”
- Line 177: replace "presents" by "is present"
It has been replaced.
Round 2
Reviewer 2 Report
The authors adequately addressed most of my concerns, and I appreciate that. The manuscript has improved a lot after revision, and now I recommend publishing. I have the following remaining questions:
1. For the added calculation of pure Fe slab in section 3.4 and Figure 6: is the supercell the same as in Fe/BTO case? Is there vacuum in between slabs, or it is a bulk calculation? Figure 6 (a) seems to indicate it is a bulk calculation without vacuum layer, but in figure 6 (b), the magnetization actually depends on layer number (total magnetization in the first and last layer is higher than others), which seems like surface effect and is unusual for bulk material.
One minor issue is that in the caption of figure 6, there is no (c), and the description of (b) seems to correspond to (c) instead.
2. Line 91: there is "??" at the end of the sentence.
Author Response
Dear Referee! Many thanks again for your careful reading!
1) Indeed, we were in a harry, and made a mistake while constructing the cell. Please, have a look again in the same Figure 6. Indeed, bulk Fe doesnt have something on the boundaries, all dependencies are constant, do not depend on the layer. Besides, the comparison with heterostructure case has revealed that in the bulk form the effect has same tendencies but much less pronounced. We have corrected the last two paragraphs in the Section 3.4 and conclusions accordingly.
The figure caption has been corrected as well.
2) The sentence has been corrected as well, there was a citation issue.